# The Impact of Serotype Cross-Protection on Vaccine Trials: DENVax as a Case Study

**DOI:** 10.3390/vaccines8040674

**Published:** 2020-11-12

**Authors:** Maíra Aguiar, Nico Stollenwerk

**Affiliations:** 1Dipartimento di Matematica, Università degli Studi di Trento, 38100 Trento, Italy; nico.biomath@gmail.com; 2Basque Center for Applied Mathematics (BCAM), E-48009 Bilbao, Basque Country, Spain; 3Ikerbasque, Basque Foundation for Science, E-48009 Bilbao, Basque Country, Spain

**Keywords:** dengue, dengue vaccine trials, vaccine efficacy, cross-protection, serotypes, serostatus, Bayesian approach

## Abstract

There is a growing public health need for effective preventive interventions against dengue, and a safe, effective and affordable dengue vaccine against the four serotypes would be a significant achievement for disease prevention and control. Two tetravalent dengue vaccines, Dengvaxia (CYD-TDV—Sanofi Pasteur) and DENVax (TAK 003—Takeda Pharmaceutical Company), have now completed phase 3 clinical trials. Although Dengvaxia resulted in serious adverse events and had to be restricted to individuals with prior dengue infections, DENVax has shown, at first glance, some encouraging results. Using the available data for the TAK 003 trial, we estimate, via the Bayesian approach, vaccine efficacy (VE) of the post-vaccination surveillance periods of 12 and 18 months. Although better measurement over a long time was expected for the second part of the post-vaccination surveillance, variation in serotype-specific efficacy needs careful consideration. Besides observing that individual serostatus prior to vaccination is determinant of DENVax vaccine efficacy, such as for Dengvaxia, we also noted, after comparing the VE estimations for 12- and 18-month periods, that vaccine efficacy is decreasing over time. The comparison of efficacies over time is informative and very important, and brings up the discussion of the role of temporary cross-immunity in dengue vaccine trials and the impact of serostatus prior to vaccination in the context of dengue fever epidemiology.

## 1. Introduction

Dengue fever is a viral mosquito-borne infection of major international public health concern, with approximately 3 billion people at risk of acquiring the infection. Caused by four antigenically related but distinct serotypes (DENV-1 to DENV-4), it is estimated that 390 million dengue infections occur every year, of which 96 million manifest symptoms with any level of disease severity [1]. Antibodies generated by exposure to any one type cross-react with other types, providing short duration cross-protective immunity. Subsequent infections by any other of the 3 heterotypic serotypes increases the risk of developing severe dengue due to an immunological process called antibody-dependent enhancement (ADE) [2,3,4,5]. Treatment of uncomplicated dengue cases is only supportive, and severe dengue cases require hospitalization. There is a compelling public health need for an effective preventive intervention against dengue. A safe, effective and affordable dengue vaccine against the four serotypes would be a significant achievement for disease prevention and control.

Two tetravalent dengue vaccines have now completed phase 3 clinical trials, and a third vaccine is currently in a phase 3 trial [6,7,8,9,10,11]. The first product, Dengvaxia (CYD-TDV), is a chimeric yellow fever tetravalent dengue vaccine developed by Sanofi Pasteur which resulted in a higher rate of hospitalized severe dengue cases when given to seronegative children, compared with age-matched seronegative controls [6,7,8]. The risks behind giving this vaccine have been discussed [12,13]. An age structured mathematical model was developed, based on Sanofi’s recommendation, and its analysis has shown a significant increase in the number of hospitalizations in a population when this vaccine was administered without population screening [14], i.e., given to both seropositive and seronegative individuals as previously suggested in [15]. Reviews in 2016–2017 identified individual serostatus prior to vaccination as determinants of Dengvaxia efficacy and adverse events [16,17], anticipating Sanofi Pasteur’s retest of the entire phase 3 serum collection announced in November 2017 [18,19]. Dengvaxia is licensed in more than 20 countries, although mass vaccination programs, initiated in the Philippines [20] and in the South of Brazil, are now suspended.

Based upon recent licensing statements (in 2019) by the E.U. and the US Food and Drug Administration (FDA), Dengvaxia administration is restricted to individuals with prior dengue infections. An issue that must be addressed is the availability of a screening test that will accurately identify dengue seropositives. This is a technical challenge, since ELISA IgG tests vary in specificity and sensitivity [21].

The second product, DENVax (TAK 003), a tetravalent chimeric vaccine developed by Takeda Pharmaceutical Company, consists of a live attenuated dengue 2 virus (DENV-2) that provides the genetic backbone for DENV 1, 3 and 4. A DENVax phase 3 trial consisting of fever-based surveillance of vaccine efficacy (VE) followed by a period of hospitalization surveillance has been completed. Case surveillance over the first 12 months after vaccination of Latin American and Asian children yielded initial encouraging results [9]. Vaccine efficacy against virologically confirmed dengue disease and hospitalization was higher and more balanced than efficacies reported for Dengvaxia [8]. DENVax achieved a 74.9% and 82.2% efficacy level [9] compared with Dengvaxia’s 52.5% and 81.9% efficacies in seronegative and seropositive children, respectively [8].

Vaccine efficacy data have now been extended to 18 months of surveillance [10] indicating that similarly to Dengvaxia, individual serostatus prior to vaccination is determinant of vaccine efficacy. Vaccine efficacy estimations for 12 and 18 months show that DENVax efficacy is decreasing over time and therefore, long-term surveillance consisting of prudent and careful observation of vaccine phase 3 recipients is required [20].

In this manuscript, DENVax vaccine efficacy for virologically confirmed dengue cases was estimated via a Bayesian approach. Vaccine efficacy measurements over time, six months apart, are compared and the variations observed in serotype-specific efficacies brings up the discussion of the role of temporary cross-immunity on dengue vaccine trials and the impact of serostatus prior to vaccination in the context of dengue fever epidemiology.

## 2. Results

Using the publicly available DENVax phase 3 trial raw data [9,10], we estimated the vaccine efficacy for virologically confirmed dengue cases via a Bayesian approach, shown in Table 1 and Table 2 for each post-vaccination surveillance period. We obtained the probability $p(k|I_{v},I_{c})$ for the vaccine efficacy *k* with infected individuals $I_{v}$ in the vaccine group and infected individuals $I_{c}$ in the control group. The statistical description of the vaccine trial data was in good agreement with the published surveillance results as described below. For detailed calculations, see Section 4. From our Bayesian analysis, vaccine efficacy estimations by serotype and serostatus are shown in Table 1 part A, using data for the first 12-month surveillance period [9]. As expected, vaccine efficacy against serotype 2, the serotype that provides the genetic “backbone” for Takeda’s DENVax, was very high independent of the individual serostatus prior to vaccination. For seropositives and seronegatives respectively, our estimations were 96.2% and 100%, in good agreement with 96.5% and 100% reported by [9]. Please note that in the seronegative vaccinated group, no dengue 2 cases were observed, but in the seropositive group there were. Vaccine efficacy against serotype 1 was observed to be slightly smaller in seronegative (estimated 79.8% as reported in [9]) than in seropositive individuals (estimated 65.9%, in good agreement with 67.2% reported in [9]). There was a negative vaccine efficacy estimated for serotype 3 in seronegative individuals (estimated −31.2% and −38.7% reported in [9]) against an intermediate vaccine efficacy in seropositives (estimated 70% and 71.3% reported in [9]), shown in Table 1 Section part A, in light blue color. Results for serotype 4 were shown to be statistically insignificant for seropositives (estimated 61.9% with 95%CI [−63.2%–91.9%) and completely “inconclusive” for seronegatives, as no cases were observed in any of the groups, vaccinated or control (see Figure 1a). Figure 2, shows the vaccine efficacy changing in time.

Table 1 Section B presents, in yellow, the overall vaccine efficacy (not by serostatus, but by serotype only), with high efficacy for serotype 2 (estimated 97.5%, in good agreement with 97.7% reported in [9]) and intermediate efficacies for serotypes 1 (estimated 73.2%, in good agreement with 73.7% reported in [9]) and serotype 3 (estimated 61.9%, in good agreement with 62.6% reported in [9]). Results for serotype 4 were statistically insignificant due to small numbers (see Figure 1b).

Table 2 is also divided in “Section A”, presenting our estimations obtained by serotype and serostatus, and in “Section B”, showing the results stratified by serotype only.

A slight decrease in vaccine efficacies was observed for the second surveillance period, with the overall vaccine efficacy estimated to be of 72.5% and the 95%-CI [65.6%, 78.1%], instead of k=79.7% and the 95%-CI [72.8%, 85.1%] estimated for the first surveillance period. Overall vaccine efficacy for specific dengue serotype 2 was estimated to be high, k=94.6% [90.3%, 97.7%], whereas vaccine efficacy for dengue serotype 1, k=69.4% [54.6%, 79.7%] and dengue serotype 3, k=47.7% [25.4%, 63.1%], were estimated to be much lower than the estimations obtained for the first 12 months of surveillance. Results for serotype 4 were still statistically insignificant, k=50.1% [−72.5%, 85.4%], including negative vaccine efficacy. Figure 3b shows our Bayesian estimates of DENVax serotype-specific vaccine efficacy.

Regarding the estimations obtained by serotype and serostatus, shown in Table 2 Section A, vaccine efficacy estimation was kept very high against DENV serotype 2, with k=93.2% and k=97.7% for seropositive and seronegative respectively. Vaccine efficacy against serotype 1 was observed to be slightly lower for seronegative, with k=66.7% [39.2%, 82.1%], than in seropositive individuals, with k=71.2% [51.8%, 83.3%]. Of concern, as shown in Table 2 section A, blue color, negative vaccine efficacy, k=−59.9% [−328.5%, 31.1%], was estimated for vaccinated seronegative individuals who were infected with serotype 3, as opposed to an intermediate efficacy, k=60.3% [40.6%, 73.2%] for seropositives. As with the other serotypes, DENV 3 efficacy declined over the following 6 months of surveillance data (see Figure 3b).

In Figure 2, we show the vaccine efficacy changing in time, decreasing from 12 to 18 months. The overall VE estimation is shown in Figure 2a. Changes in VE for seronegative individuals (see Figure 2b) and for seropositive individuals (see Figure 2c) as also shown. Although VE for seronegative individuals are lower, as also observed for Dengvaxia, the trend of VE lowering over time is also observed for seropositive individuals. Vaccine efficacy estimations for serotype 4 were kept statistically insignificant during the 18 months of surveillance. The small number of cases contained a suggestion of negative vaccine efficacy for vaccinated seropositive children. The possibility exists that efficacy against DENV 4 may follow the pattern of serotype 3 as case numbers increase.

Curiously, for both trials combined, the Asian-Pacific and the Latin American, Sanofi Pasteur’s Dengvaxia produced higher vaccine efficacies against serotypes 3 and 4 (over 70%, see Figure 3a) than to serotypes 1 and 2 (around 40%) [22] while the pattern of Takeda DENVax efficacy has been the opposite, high overall vaccine efficacies for serotypes 1 and 2 and moderate to low vaccine efficacy for serotypes 3 and 4 (see Figure 3b)).

## 3. Discussion

DENVax vaccine efficacy against virologically confirmed dengue disease and hospitalization were shown to be higher than efficacies reported for the Sanofi Pasteur tetravalent dengue vaccine, Dengvaxia, with a more balanced efficacy in seronegatives and seropositives. Although Takeda’s DENVax vaccine efficacy was estimated to be of 74.9% for participants who were seronegative and 82.2% for participants who were seropositive at baseline (see Table 1, Section B), the Sanofi Pasteur’s Dengvaxia has shown a 52.5% and 81.9% vaccine efficacies in seronegative and seropositive individuals respectively [8]. Although more investigation is needed to identify the reason for DENVax presenting a more balanced VE by serotype than Dengvaxia, it is important to acknowledge that the trial design for the Takeda’ vaccine has enabled an immediate and detailed assessment of vaccine efficacy of subgroups.

Using the new DENVax phase 3 trial data, vaccine efficacy measurements over time, six months apart, are compared. Significant variations in vaccine efficacy are observed, with estimations decreasing over time in serotype-specific analysis. It is important to note that these results obtained within 12–18 months post-vaccination periods are still within the well-established period of cross-protective immunity against clinical disease that follows a first dengue infection [23,24]. This trend suggests that early DENVax protective efficacy may be attributed to cross-protection and therefore may continue to decline over time.

Of concern, negative vaccine efficacy was estimated for vaccinated seronegative individuals who were infected with serotype 3, as opposed to an intermediate efficacy for seropositives. The possibility exists that efficacy against DENV 4 may follow the pattern of serotype 3 as case numbers increase, therefore, efficacy and safety conclusions require long-term surveillance.

Variations in serotype-specific efficacies of DENVax are concerning when it is recalled that for Dengvaxia high rates of hospitalization of vaccinated young children, mis-interpreted as vaccine failure, were resolved long after the vaccine was widely licensed and administered in millions of children. For Dengvaxia, initial low vaccine efficacy was shown to include vaccine adverse events and a significant incidence of hospitalization for severe dengue in vaccinated seronegative children. With respect to DENVax we urge the vaccine community to adopt the stance of “watch and wait” because VE is not a static measure but appears to be time dependent due to, for example, temporary cross-protection. Final conclusions can only be drawn after some year of evaluation. It is too soon to understand the behavior of this vaccine in individuals of differing immunological serostatus or age.

Besides observing that individual serostatus prior to vaccination is determinant of DENVax vaccine efficacy, we also noted that the DENVax efficacy is decreasing over time. The comparison of efficacies over time is informative and very important since it brings up the discussion of the role of temporary cross-immunity on dengue vaccine trials and the impact of serostatus prior to vaccination in the context of dengue fever epidemiology.

With respect to DENVax, long-term surveillance consisting of prudent and careful observation of vaccine phase 3 recipients is required. Careful design of vaccine policies is urgently needed and recommendations concerning the use of dengue vaccines should consider a better measurement of vaccine efficacy over time and safety through enhanced phase 4 surveillance.

## 4. Materials and Methods

Vaccine trials can be modeled using a previously investigated simple epidemiological process, the linear infection model [25], which can be solved analytically in all aspects, and thus serve as a test model for many further aspects, like parameter estimation, model comparison, or analytics of approximation schemes. [26,27,28].

In this section, we describe the methodology used to estimate DENVax vaccine efficacy via a Bayesian approach, using the data from Takeda’s dengue vaccine trial in Latin America and Asia [9,10].

### 4.1. Linear Infection Model

The basic assumptions for modeling a vaccine trial are exactly the ones described by the linear infection model, which has the following reaction scheme
(1)$$\begin{array}{ccc} S+I^{*} & \overset{\beta }{⟶} & I+I^{*} \end{array}$$
for infected *I* and susceptibles S=N−I, with population size *N*, and infection rate β as the only possible transition. The underlying model hypothesis is that infection can be acquired from outside the considered trial population of size *N* by meeting a constant number of infected individuals $I^{*}$ from a much larger background population. The master equation for the probability p(I,t) is
(2)$$\begin{array}{ccc} \frac{d}{dt}p\left(I,t\right) & = & \frac{\beta }{N}I^{*}\cdot \left(N-\left(I-1\right)\right)p\left(I-1,t\right)-\frac{\beta }{N}I^{*}\cdot \left(N-I\right)p\left(I,t\right) \end{array}$$
which can be solved using the characteristic function
(3)$$\langle e^{i\kappa I}\rangle :=\sum_{I=0}^{N}e^{i\kappa I}\cdot p\left(I,t\right)=:g\left(\kappa ,t\right)$$
with $\beta^{*}:=\left(\beta /N\right)I^{*}$ as the external infection level, and initial condition $p\left(I,t_{0}\right)=\delta_{I,I_{0}}$, as described in more detail in [26]. The solution of the master equation is given by
p(I,t)=N−I0I−I0e−β*(t−t0)N−I1−e−β*(t−t0)I−I0.

Due to the special initial conditions of having exactly $I_{0}$ infected at time $t_{0}$ this solution is at the same time the transition probability $p(I,t|I_{0},t_{0})$, needed to construct the likelihood function for parameter estimation of data vector $\underline{I}=\left(I_{0},I_{1},...,I_{n}\right)$. See [26] for further discussion on this point.

### 4.2. Modeling Vaccine Trials with the Linear Infection Model

For modeling a vaccine trial, we have the special case of one data point each for the processes of the “control group” and the “vaccine group”. For the control group we have the scheme
(4)$$\begin{array}{ccc} S_{c}+I^{*} & \overset{\beta }{⟶} & I_{c}+I^{*} \end{array}$$
and with initially $N_{c}$ participants in the control group, we have after a time interval $T:=t-t_{0}$ (here T=12 respectively 18 months for the DENVax vaccine trial [9,10]) the solution of the process given by the probability of having $I_{c}$ disease cases in the control group
p(Ic,T)=NcIce−β*TNc−Ic1−e−β*TIc
with parameter $\theta_{\beta }:=e^{-\beta^{*}T}$ incorporating the effects of external infection levels over the time period *T*. The parameter $\theta_{\beta }$ is also the probability not to become infected over the considered time interval. For parameter estimation purposes, the solution of the master equation $p(I_{c},T)$ is also the likelihood of the single model parameter $\theta_{\beta }$ of the control group process, hence $p\left(I_{c},T\right)=:L\left(\theta_{\beta }\right)=p\left(I_{c}|\theta_{\beta }\right)$, with the last notation as used in the Bayesian statistical framework [25,26,27].

For the vaccine group we have the same epidemiological process, but with (hopefully) reduced infectivity c·β instead of β, or with vaccine efficacy k:=1−c,
(5)$$\begin{array}{ccc} S_{v}+I^{*} & \overset{(1-k)\beta }{⟶} & I_{v}+I^{*} \end{array}$$
and solution for $N_{v}$ participants in the vaccine group to find $I_{v}$ infected still (despite the vaccination effort)
p(Iv,T)=NvIve−(1−k)β*TNv−Iv1−e−(1−k)β*TIv
now with parameter $\theta_{k}:=e^{-\left(1-k\right)\beta^{*}T}={\left(e^{-\beta^{*}T}\right)}^{1-k}=\theta_{\beta }^{1-k}$. The master equation solution is now again to be interpreted as a likelihood function $p\left(I_{v},T\right):=L\left(\theta_{k}\right)=p\left(I_{v}|\theta_{k}\right)$ or in terms of the already estimated $\theta_{\beta }$ we have as well $p\left(I_{v},T\right)=p\left(I_{v}|k,\theta_{\beta }\right)$. Hence we must estimate the parameter of external infection level $\theta_{\beta }$ from the data of the control group, and then estimate the vaccine efficacy *k* from the information of the vaccine group. Via the Bayesian approach we obtain explicitly a probability for the vaccine efficacy based on the empirical data from the vaccine trial, $p(k|I_{v},I_{c})$, where insecurity of the intermediate parameter of external infection level $\theta_{\beta }$ is taken into account by marginalizing over this parameter.

Previously, we have given from the linear infection model the statistical description of the vaccine trial data via the Bayesian approach in very good agreement with the results in [9,10].

## 5. Bayesian Analysis

From the above-mentioned likelihood functions we can immediately calculate posterior distributions for the parameters given the data, once the priors for the parameters are chosen. For details especially concerning the linear infection model see [25], and then also [26,27].

In detail, from the likelihood function of $\theta_{\beta }$ for the control group of the vaccine trial $L\left(\theta_{\beta }\right)\;=\;p\left(I_{c}|\theta_{\beta }\right)$ which has the form of a beta-distribution, we can obtain the posterior distribution $p(\theta_{\beta }|I_{c})$ by using for the prior a conjugate form of a beta-distribution with parameters *a* and *b* as
(6)$$p\left(\theta_{\beta }|I_{c}\right)=\frac{p(I_{c}|\theta_{\beta })}{p(I_{c})}\;p\left(\theta_{\beta }\right)$$
with a normalization constant, also called “evidence” in the context of model comparison [27] and further references cited there, $p\left(I_{c}\right)=\int p\left(I_{c}|\theta_{\beta }\right)\cdot p\left(\theta_{\beta }\right)\;d\theta_{\beta }$. Then the prior can be given as
(7)$$p\left(\theta_{\beta }\right)=\frac{\Gamma (a+b)}{\Gamma (a)\cdot \Gamma (b)}\;\theta_{\beta }^{a-1}\cdot {\left(1-\theta_{\beta }\right)}^{b-1}$$
with parameters *a* and *b* to be chosen to be much smaller than the respective exponents in the likelihood function.

We chose a=b=0.5 giving a dish shaped distribution with lowest values around $\theta_{\beta }=0.5$, hence in no way resembling any one-humped distribution as expected for the posterior. After some calculations, see [25] for any details, we obtain the posterior distribution, the probability density to find the background infection level conditioned on the control group outcome data, as
(8)$$p\left(\theta_{\beta }|I_{c}\right)=\frac{\Gamma (a+k_{2}+b+k_{3})}{\Gamma \left(a+k_{2}\right)\cdot \Gamma \left(b+k_{3}\right)}\;\theta_{\beta }^{a+k_{2}-1}\cdot {\left(1-\theta_{\beta }\right)}^{b+k_{3}-1}$$
with $k_{2}=N_{c}-I_{c}$ and $k_{3}=I_{c}$ as data dependent parameters. Please note that another constant $k_{1}$ is used in likelihood maximization and cancels out in the Bayesian approach [28].

To calculate the finally sought quantity $p(k|I_{v},I_{c})$ we also need to obtain from the likelihood $L\left(k\right)=p\left(I_{v}|k,\theta_{\beta }\right)$ for the vaccine group the Bayesian posterior $p(k|I_{v},\theta_{\beta })$ with
(9)$$p\left(k|I_{v},\theta_{\beta }\right)=p\left(\theta_{k}|I_{v}\right)\cdot \frac{d\theta_{k}}{dk}$$
with $\theta_{k}=\theta_{\beta }^{1-k}$ as defined above. Again, after some calculations using the conjugate prior
(10)$$p\left(\theta_{k}\right)=\frac{\Gamma (a_{v}+b_{v})}{\Gamma \left(a_{v}\right)\cdot \Gamma \left(b_{v}\right)}\;\theta_{k}^{a_{v}-1}\cdot {\left(1-\theta_{k}\right)}^{b_{v}-1}$$
being as uninformed as possible by choosing $a_{v}=b_{v}=0.5$ again, we obtain the result
(11)$$\begin{array}{ccc} p(k|I_{v},\theta_{\beta }) & = & \frac{\Gamma (a_{v}+k_{v2}+b_{v}+k_{v3})}{\Gamma \left(a_{v}+k_{v2}\right)\cdot \Gamma \left(b_{v}+k_{v3}\right)}\;\theta_{k}^{a_{v}+k_{v2}-1}\cdot {\left(1-\theta_{k}\right)}^{b_{v}+k_{v3}-1}\cdot \left(\ln\left(\frac{1}{\theta_{\beta }}\right)\cdot \theta_{\beta }^{1-k}\right) \end{array}$$
with $k_{v2}=N_{v}-I_{v}$ and $k_{v3}=I_{v}$ as defined above.

However, to calculate the finally sought $p(k|I_{v},I_{c})$, using the previously calculated posteriors $p(\theta_{\beta }|I_{c})$ and $p(k|I_{v},\theta_{\beta })$, we need to integrate over the parameter $\theta_{\beta }$ whose insecurity has not been taken into account by our considerations up to now. Hence we calculate $p(k|I_{v},I_{c})$ as the marginal probability distribution of the joint probability distribution $p\left(k,\theta_{\beta }|I_{v},I_{c}\right)=p\left(k|I_{v},\theta_{\beta }\right)\cdot p\left(\theta_{\beta }|I_{c}\right)$, namely
(12)$$p\left(k|I_{v},I_{c}\right)=\int_{0}^{1}p\left(k|I_{v},\theta_{\beta }\right)\cdot p\left(\theta_{\beta }|I_{c}\right)\;d\theta_{\beta }$$
with the final result plotted in Figure 1, and its cumulative distribution function $P(k|I_{v},I_{c})$in order to read off the confidence intervals around the median as the best estimator in the Bayesian approach.

From the data which generates Figure 1a for example (the others follow accordingly), we obtain from the median of the marginalized posterior $P(k_{0.5}|I_{v},I_{c})=0.5$ the Bayesian estimate of the vaccine efficacy $k_{0.5}=81.4\%$ for overall seropositives and the 95%-confidence interval from the 0.025 and 0.975 quantiles, hence $P(k_{0.025}|I_{v},I_{c})=0.025$ for the lower bound $k_{0.025}=73.6\%$ and $P(k_{0.975}|I_{v},I_{c})=0.975$ for the upper bound $k_{0.975}=87.1\%$. This is in good agreement with the values given in [9] (with xx% (95%-CI: xx–xx)), given the somewhat arbitrary Bayesian priors with their parameters *a*, *b*, $a_{v}$ and $b_{v}$.

In total we have given from the linear infection model the statistical description of the vaccine trial data via the Bayesian approach in very good agreement with the results in [9,10] and in addition the graphical representation of the posteriors and Figure 1, Figure 2 and Figure 3, indicating where the bulks of the probabilities are concentrated.

## Figures and Tables

**Figure 1 vaccines-08-00674-f001:**
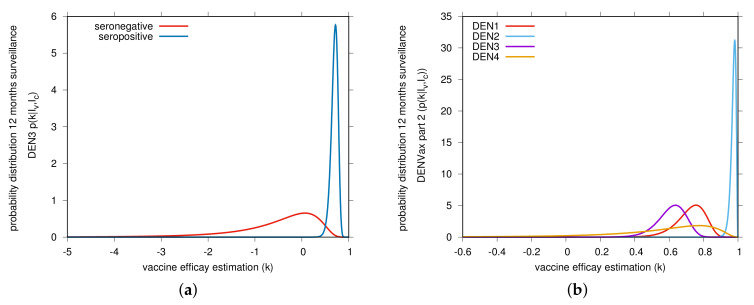
Bayesian estimate of DENVax vaccine efficacies (VE) against virologically confirmed dengue for the first 12-month surveillance (part 1). In (**a**) DEN3 serotype-specific vaccine efficacy by serostatus. Red and blue curves show our estimates for the seronegative and seropositive individuals, respectively. In (**b**) serotype-specific VE estimations. The raw data used to estimate the distribution by dengue serotype for individuals aged 4–16 years old was obtained in [9] and are shown in Table 1. High vaccine efficacy for dengue serotype 2 and intermediate to low vaccine efficacy for the other serotypes are observed.

**Figure 2 vaccines-08-00674-f002:**
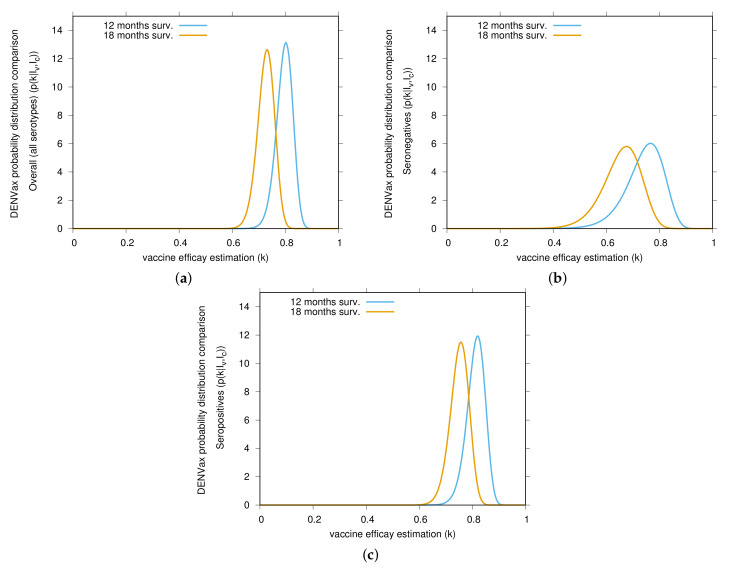
Bayesian estimates comparison of DENVax vaccine efficacies (VE) against virologically confirmed dengue cases. Yellow and light blue curves show the estimates for the first 12 months (part 1) and the 18 months (part 2) surveillance, respectively. In (**a**) overall VE estimations for all serotypes, in (**b**) VE estimation for seronegative individuals and in (**c**) VE estimations for seropositive individuals. Data from Table 1 and Table 2, in good agreement with results reported in [9,10], are used to estimate the distribution by dengue serotype and serostatus for individuals aged 4–16 years old. Vaccine efficacy is decreasing over time.

**Figure 3 vaccines-08-00674-f003:**
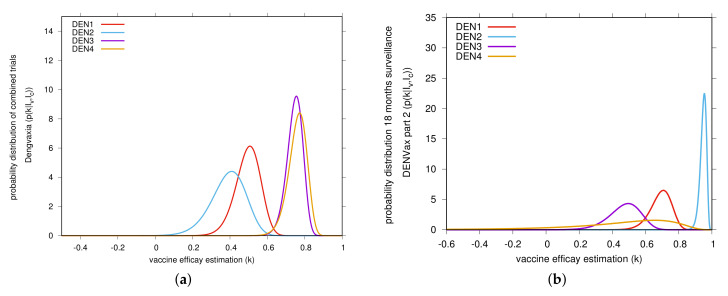
Bayesian estimate of vaccine efficacies (VE) against virologically confirmed dengue. In (**a**) serotype-specific VE for Dengvaxia, Sanofi Pasteur. The raw data used for VE estimation by dengue serotype for individuals aged 9 years and older was obtained in [8]. In (**b**) serotype-specific VE for DENVax, Takeda. The raw data used to estimate the VE by dengue serotype for individuals aged 4–16 years old was obtained in [10] and are shown in Table 2.

**Table 1 vaccines-08-00674-t001:** Compilation of vaccine efficacies estimation for Takeda’s DENVax vaccine phase 3 trial post-vaccination surveillance period part 1 (12 months). Section A shows the vaccine efficacies by serostatus and serotype, and Section B shows the overall vaccine efficacy by serotype. Highlighting problems observed for serotypes 3 (blue) and 4 (grey) are indicated. The raw data used for the Bayesian analysis are available in [9].

Part 1 Efficacy Data of the TAK-003 Phase 3 Trial									
section A							section B		
	Seropositive at baseline (82.2%)			Seronegative at baseline (74.9%)			Overall (seropositive and seronegative)		
							vaccine efficacy		
Dengue Serotype	Vaccinated	Control	Estimated vaccine	Vaccinated	Control	Estimated Vaccine	Vaccinated	Control	Estimated vaccine
	(n=9167)	(n=4589)	efficacy and 95%	(n=3531)	(n=1726)	efficacy and 95%	(n= 12,700)	(n=6316)	efficacy and 95%
	Dengue cases	Dengue cases	Confidence Interval	Dengue cases	Dengue cases	Confidence Interval	Dengue cases	Dengue cases	Confidence Interval
ALL	41	110	81.4%	20	39	74.8%	61	149	79.7%
			[73.6%,87.1%]			[57.4%,85.4%]			[72.8%,85.1%]
DEN1	7	17	78.9%	9	13	65.9%	16	30	73.2%
			[51.8%,91.4%]			[22.2%,85.8%]			[52.2%,85.4%]
DEN2	3	42	96.2%	0	22	100%	3	64	97.5%
			[89.9%,98.8%]						[93.6%,99.3%]
DEN3	28	47	70.0%	11	4	−31.2%	39	51	61.9%
			[52.6%,81.4%]			[−353.2%,53.8%]			[42.4%,75.8%]
DEN4	3	4	61.9%	0	0	inconclusive	3	4	61.9%
			[−63.2%,91.9%]						[−62.4%,91.9%]

**Table 2 vaccines-08-00674-t002:** Compilation of vaccine efficacies estimation for Takeda’s DENVax vaccine phase 3 trial post-vaccination surveillance period part 2 (18 months). Section A shows the vaccine efficacies by serostatus and serotype, and Section B shows the overall vaccine efficacy by serotype. Highlighting problems observed for serotypes 3 (blue) and 4 (grey) are indicated. The raw data used for the Bayesian analysis are available in [10].

Part 2 Efficacy Data of the TAK-003 Phase 3 Trial									
section A							section B		
	Seropositive at baseline (82.2%)			Seronegative at baseline (74.9%)			Overall (seropositive and seronegative)		
							vaccine efficacy		
Dengue Serotype	Vaccinated	Control	Estimated vaccine	Vaccinated	Control	Estimated Vaccine	Vaccinated	Control	Estimated vaccine
	(n=9167)	(n=4589)	efficacy and 95%	(n=3531)	(n=1726)	efficacy and 95%	(n= 12,700)	(n=6316)	efficacy and 95%
	Dengue cases	Dengue cases	Confidence Interval	Dengue cases	Dengue cases	Confidence Interval	Dengue cases	Dengue cases	Confidence Interval
ALL	75	150	75.8%	39	56	66.1%	114	206	72.5%
			[67.2%,81.0%]			[48.9%,77.3%]			[65.6%,78.1%]
DEN1	21	37	71.2%	17	25	66.7%	38	62	69.4%
			[51.8%,83.3%]			[39.2%,82.1%]			[54.6%,79.7%]
DEN2	7	54	93.2%	1	26	97.7%	8	80	94.6%
			[91.1%,97.1%]			[90.7%,99.7%]			[90.3%,97.7%]
DEN3	43	54	60.3%	20	6	−59.9%	63	60	47.7%
			[40.6%,73.2%]			[−328.5%,31.1%]			[25.4%,63.1%]
DEN4	4	5	59.5%	1	0	inconclusive	5	5	50.1%
			[−47.2%,89.4%]						[−72.5%,85.4%]

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
