# Peer review of "The Impact of Serotype Cross-Protection on Vaccine Trials: DENVax as a Case Study"

_vaccines, 2020, doi:10.3390/vaccines8040674_

Round 1

Reviewer 1 Report

As stated by the authors in their introduction, "In this manuscript, DENVax vaccine efficacy for virologically confirmed dengue cases are estimated via a Bayesian approach". I was then expecting to read the conclusions of the bayesian approach in predicting vaccine efficacy.

Unfortunately, the Results section mostly deals with basics statistics on vaccine efficacy at 12 months and 18 months post-vaccination for the DENVax candidate. I still do not understand why the authors do not even comment their own results on bayesian predictions of vaccine efficacy, while the 3 figures are devoted to probability distributions and the entire Material and Methods section motivates the use of bayesian approaches and the computation of probability distributions. The manuscript ends up being a description of the results in Tables 1 and 2, and nothing more.

Noticeably, the authors do not manage to make a single correct reference to the figures. They always refer to Figure 2, even when they should refer to Figures 1 or 3, and their only reference to Figure 1 (line 111) is wrong, it should be Figure 2. It seems the manuscript has been written very quickly, this is highlighted by bibliographic references with 2 numbers (4, 5 and 10 for instance).

Additionally, twice in the manuscript the authors comment on the possibility that "efficacy against DENV 4 may follow the pattern of serotype 3 as case numbers increase." (lines 118-119136-137). This is a purely gratuitous assertion that should not appear in this manuscript, unless the authors have some scientific clues about it, which they did not show in the current paper.

Finally, in the description of their bayesian approach, the authors should at least motivate and comment on the choice of a = b = 1/2. Since they do not describe any of their results related to this approach it may be useless that I make a comment on that, but scientifically the question deserves to be addressed. What is the sensitivity of the results of the choice of a and b? Did you perform a sensitivity analysis?

Reviewer 2 Report

This is a timely and innovative approach to an age-old problem. The significance is adequate to merit publication and it aligns well with the scope of this journal. The quality of presentation is sound and the methodology lends novelty to the paper. The methods, results and presentation are adequate to justify the conclusions. 

Reviewer 3 Report

Based on the publicly available DENVax phase 3 trial data, the authors study the efficacy of the vaccine using a Bayesian approach. The paper is well structureed and reads well. The methodology appears sound and the results are presented in a consise way and summarized in table 2.

A minor comment: It would helpful though if the authors would show also the upper and lower bounds of the vaccine efficacy in Figures 1,2,3.

Reviewer 4 Report

The English of the article is  OK but many sentences are too long and carry too many considerations and subjects. Please try to make 2 or 3 short sentences for each long one that expresses 2 or 3 ideas.

*Also:

Abstract, l.5: it should read: "and had to be restricted.."

Abstract, l.13-14: these two lines are useless: could be suppressed

Intro, l.18:write "of" in place of "a" ("an infection of major international"...)

l.21: "which" should be better than "and" ("wich resulted in a higher rate...")

p2, l.35 "in"  instead of "on" (" an increase in the number"...)

l.54: "a 74.9% and 83.2% efficacy level [9]compared with..."

l.61: "vaccine efficacy against virologically confirmed cases was..."

p3, l.90: should read: "statistically", not " statically"

Discussion, l. 129-130:vaccine efficacy was 74.9% for seronegative participants versus 82.2% for seropositive ones..."' 

l. 122: cross out "as reported in", it should be suppressed

l. 133-134: . These two lines ought to be suppressed!

l. 141-143: these two lines are hard to understand: their meaning is obscure! Please explain!

Overall, the discussion reads more like a summary than a discussion. Is it clear that one vaccine appears to be superior to the other? Why is it so? What could be the explanation for their difference? Even if only tentative, a discussion on the subject would render the article much more appealing than the 3 pages of equations in its Materials and Methods section。

Round 2

Reviewer 1 Report

The revised version is much clearer. The results are now convincing, so I have no reserve regarding publication of the manuscript.

I have to mention that the manuscript is still full of typos, I don't think my role is to evaluate such things, so I let the journal staff or the Editor deal with that.

Author Response

Dear Reviewer, Thank you for your positive comments.

The manuscript was revised and updated (Nov4) by a native English speaker and the typos were corrected.

With our best regards,

The Authors